# Experimental Study on the Heat Transfer Performance of Pump-Assisted Capillary Phase-Change Loop

**Xiaoping Yang [1], Gaoxiang Wang [1], Cancan Zhang [2], Jie Liu [1] and Jinjia Wei [1,3,***

[1] College of Chemical Engineering and Technology, Xi'an Jiaotong University, Xi'an 710049, China; yxping@xjtu.edu.cn (X.Y.); wgxxjhgxy@stu.xjtu.edu.cn (G.W.); liujie0101@stu.xjtu.edu.cn (J.L.)

[2] Key Laboratory of Enhanced Heat Transfer and Energy Conservation, Ministry of Education of China, College of Environmental and Energy Engineering, Beijing University of Technology, Beijing 100124, China; zcc@bjut.edu.cn

[3] State Key Laboratory of Multiphase Flow in Power Engineering, Xi'an Jiaotong University, Xi'an 710049, China

* Correspondence: jjwei@xjtu.edu.cn; Tel.: +86-029-82664375

**Featured Application: The pump-assisted capillary phase-change loop overcomes the two-phase flow instability of traditional boiling heat dissipation technologies. Thus, it shows prospects for use in the heat dissipation of electronics devices and batteries on the ground and in space in applications such as datacenter cooling, and thermal management of batteries or spacecraft.**

**Abstract:** To overcome the two-phase flow instability of traditional boiling heat dissipation technologies, a porous wick was used for liquid-vapor isolation, achieving efficient and stable boiling heat dissipation. A pump-assisted capillary phase-change loop with methanol as the working medium was established to study the effect of liquid-vapor pressure difference and heating power on its start-up and steady-state characteristics. The results indicated that the evaporator undergoes four heat transfer modes, including flooded, partially flooded, thin-film evaporation, and overheating. The thin-film evaporation mode was the most efficient with the shortest start-up period. In addition, heat transfer modes were determined by the liquid-vapor pressure difference and power. The heat transfer coefficient significantly improved and the thermal resistance was reduced by increasing liquid-vapor pressure as long as it did not exceed 8 kPa. However, when the liquid-vapor pressure exceeded 8 kPa, its influence on the heat transfer coefficient weakened. In addition, a two-dimensional heat transfer mode distribution diagram concerning both liquid-vapor pressure difference and power was drawn after a large number of experiments. During an engineering application, the liquid-vapor pressure difference can be controlled to maintain efficient thin-film evaporation in order to achieve the optimum heat dissipation effect.

**Keywords:** liquid cooling; phase-change loop; pressure difference; heat transfer enhancement

## 1. Introduction

The heat transfer coefficient of phase-change heat dissipation is 1–2 orders of magnitude higher than single-phase convection heat dissipation. Hence, phase-change heat dissipation technology can be adopted to save pumping power and ensure a more compact heat dissipation structure [1,2]. For this reason, phase-change heat dissipation technology has attracted extensive attention in the field of thermal management of electronic devices under the conditions of normal gravity on the ground and microgravity in space. Due to its high efficiency and compactness, phase-change heat dissipation technology has been applied to the cooling of electronic devices on spacecraft (such as satellites, spaceships, and space stations), such as passive loop heat pipe (LHP) [3], capillary pumped loop (CPL) [4], and active pump-driven two-phase loop [5]. Phase-change heat dissipation displays high efficiency under normal gravity conditions on the ground, enabling it to be widely applied

in the fields of high heat flux chip cooling and battery thermal management, with the pump-driven micro/mini-channel cooling plate as its main form [6–11]. However, the phase-change cooling system may become unstable in pressure, temperature, and flow during operation, which may not only reduces critical heat flux but also causes problems such as vibration, noise, and local hot spots [12,13]. Therefore, overcoming two-phase flow instability in the phase-change cooling system has become a focus of research.

For years, many scholars have carried out research on the mechanism of two-phase flow instability in a phase-change cooling system from both macro-scale [14,15] and micro-scale channels [16–19]. Ding et al. [14] carried out an experimental study on the pressure pulsation caused by flow boiling in a horizontal pipe with a diameter of 10.9 mm and found three types of pulsation: pressure drop type, density wave type, and thermal type. According to the dynamic pressure test results, the pressure pulsation amplitude of pressure-drop type instability may reach 160 kPa with a frequency of less than 3 Hz, which may easily cause pipeline resonance. With significant advantages such as strong heat transfer capacity, compact structure, and easy integration, micro-channel flow boiling is promising in the thermal management of high heat flux electronic devices [20,21]. However, the applications of two-phase flow are limited due to its instability. Wu and Cheng [16] carried out synchronous visualization research. They found that the flow patterns of micro-channel flow boiling are closely related to temperature, pressure, and flow pulsation. Specifically, the alternate flow of liquid/vapor/liquid-vapor under high heat flux and low mass flux may cause strong pulsation, with a pressure fluctuation amplitude of up to 44 kPa and a temperature fluctuation of up to 229 °C. This shows there are many types of two-phase flow instability in the phase-change cooling system, which are closely related to the transition of flow patterns. Flow pattern is a special physical phenomenon of vapor phase dispersion in the liquid phase. The compressibility of liquid-vapor two-phase flow and the evolution of vapor volume fraction may cause fluctuations in both upstream and downstream flow field parameters (pressure, temperature, and flow rate).

The surface micro/nanostructure can be used in the micro-channel to increase the nucleate site density, promote the nucleation of bubbles at low wall superheat, and restrain boiling delay, thus not only eliminating the two-phase instability but also enhancing boiling heat transfer [22–24]. Some scholars have put forward methods for active suppression of micro-channel flow boiling instability, such as increasing the inlet liquid pressure to prevent backflow [25], using a subcooled liquid jet to prevent bubbles from growing [26,27], etc. However, the micro/nanostructure is complicated to prepare and cannot be used for active control of instability and the heat transfer process. Active regulating methods may cause a large flow pressure drop and extra pump work.

The root cause of two-phase flow instability in the phase-change cooling system is the various liquid-vapor two-phase flow phenomena in the phase-change process. In this paper, the idea of isolating the vapor phase from the liquid phase based on the above analysis is proposed. The specific method is as follows: a porous capillary wick is set between the evaporation surface and the liquid channel so that liquid can be supplied to the evaporation surface through the liquid permeability of the capillary wick. The gas produced by phase-change can be isolated through the vapor isolation characteristics of the capillary wick, which prevents the formation of liquid-vapor two-phase flow, thus solving the two-phase instability of a phase-change cooling system. By establishing a pump-assisted capillary phase-change loop system with methanol as the working medium, this paper studies the effect of liquid-vapor pressure difference on both sides of the capillary wick on the start-up characteristics and steady-state heat transfer characteristics. The proposed system was found to be good for overcoming two-phase flow instability. The heat transfer mode of the evaporator can be effectively controlled by controlling the liquid-vapor pressure difference to enable the evaporator to form efficient thin-film evaporation heat transfer and significantly enhance heat transfer.

## 2. Experimental System

### 2.1. Experimental Apparatus

Figure 1 presents a system sketch of a pump-assisted capillary phase-change loop. The phase-change loop consists of an evaporator, liquid tank, diaphragm pump, vapor/liquid pipelines, simulated heating system, valve, temperature/pressure sensors, data acquisition system, etc. The working medium methanol was stored in the liquid tank, and its temperature controlled by an industrial cooler. The diaphragm pump delivered the liquid working medium through the liquid line to the upper compensation chamber of the evaporator. A portion of the liquid entered the baseplate of the evaporator for heat dissipation through the capillary wick, while the remaining liquid flowed out the other end of the compensation chamber to the liquid tank. After entering the baseplate of the evaporator, the liquid absorbed heat and phase-change occurred, and then it flowed out of the gas pipeline to the liquid tank for condensation. The remaining liquid and vapor flowed into the upper space of the liquid tank. Pressure measuring points were set at the outlets of both the compensation chamber and the vapor collector to monitor both the pressure of subcooled liquid $p_L$ and the pressure of hot fluid $p_V$. High precision regulating valves were installed on the liquid lines at both the inlet and outlet of the compensation chamber to regulate the liquid pressure of the compensation chamber, which made it possible to control the liquid-vapor pressure difference between both sides of the capillary wick to $\Delta p_{LV} = p_L - p_V$. The thermophysical properties of the working medium methanol are shown in Table 1.

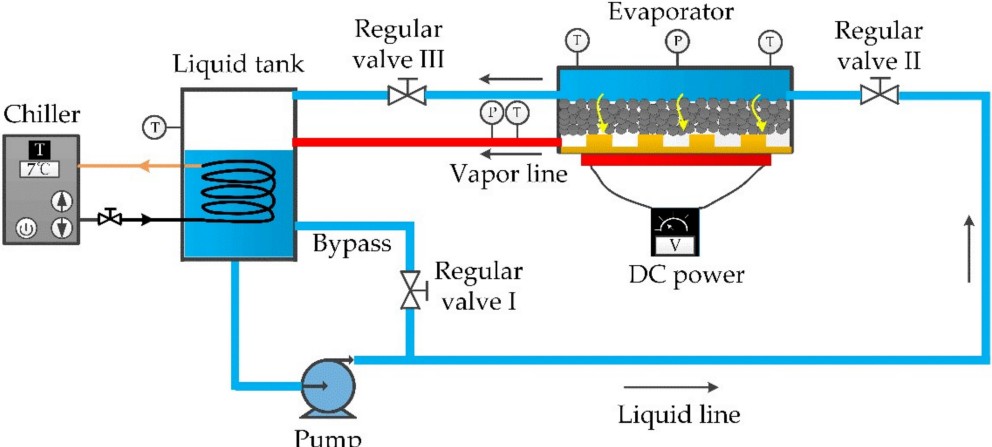

**Figure 1.** Schematic diagram of the pump-assisted capillary phase-change loop.

**Table 1.** Thermophysical properties of methanol at room temperature and pressure.

| Parameter | Unit | Value |
|---|---|---|
| Boiling point | °C | 64.7 |
| Freezing | °C | −97.8 |
| Kinematic viscosity | mPa·s | 0.59 |
| Latent heat | kJ/kg | 1109 |
| Surface tension | mN/m | 22.6 |
| Specific heat capacity | kJ/kg·K | 2.51 |
| Density | kg/m³ | 792 |

During the experimental procedure, the cooler was started first and the pump was started as soon as the liquid tank temperature reached 10 °C. Secondly, the liquid pressure $\Delta p_{LV}$ was adjusted to a certain value by regulating valves and the heating system was started to generate joule heat on the heating plate. Finally, the heating power was gradually increased when the baseplate temperature reached a steady state. When the heating plate

temperature reached the limit of the material (180 °C in the present experiment), one group of experiments ended. The next group of experiments was started by adjusting the liquid pressure $p_L$ to another certain value.

The evaporator is a core part of the pump-assisted capillary phase-change loop, including an upper cover plate, capillary wick, copper baseplate, heating plate, and support plate reaching from top to bottom, as shown in Figure 2. The compensation chamber, liquid inlet/outlet channels, and vapor collector were set on the upper baseplate, while pressure measuring points were set on the sides of the compensation chamber to monitor fluid pressure. The capillary wick was located between the upper cover plate and copper baseplate and embedded directly below the compensation chamber. The copper baseplate was close to the capillary wick, with a rubber ring used to achieve a fluid seal. The heating plate was attached to the copper baseplate using thermally conductive silicone grease. The main dimensions of the evaporator are shown in Table 2.

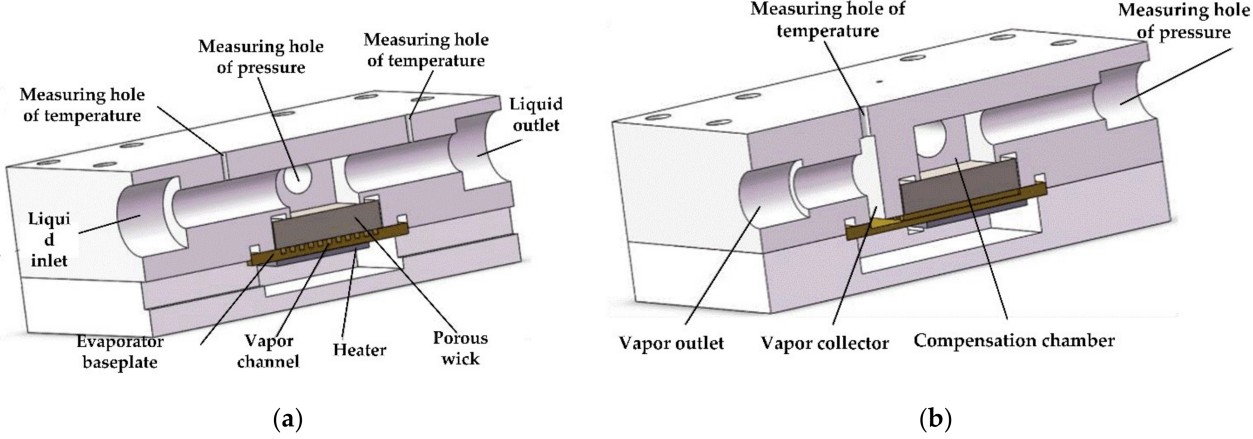

(**a**)    (**b**)

**Figure 2.** Three-dimensional diagram of the evaporator: (**a**) front view of cross-section drawn; (**b**) side view of cross-section drawn.

**Table 2.** Main dimensions and test conditions in the experiments.

| Components | Parts | Material | Dimensions |
|---|---|---|---|
| Evaporator | Baseplate | Copper | Overall size: $36 \times 30 \times 2$ mm$^3$<br>Vapor channel size: $1 \times 1 \times 23$ mm$^3$<br>Vapor channel number: 11 |
| | Wick | Sintered nickel powder | Overall size: $23 \times 23 \times 5$ mm$^3$<br>Particle size: 20 μm<br>Pore-forming agent content: 15 wt%<br>Porosity: 33%<br>Permeability: $2.99 \times 10^{-11}$<br>Average pore size: 2 μm |
| | Compensation chamber | Polycarbonate | $60 \times 20 \times 12$ mm$^3$ (inner) |
| Heater | Substrate | Silicon | Active heating area: $20 \times 20$ mm$^2$<br>Resistance: 50 Ω |

The capillary wick was rectangular, with dimensions of $23 \times 23 \times 5$ mm$^3$. It was made by sintering spherical nickel powder particles with a particle size of 20 μm using the process of cold press sintering at the sintering temperature of 840 °C. In order to improve the porosity of the capillary wick, $Na_2CO_3$ particles with a mass fraction of 15% were added as pore formers to the nickel powder particles. After sintering, the capillary wick was obtained through ultrasonic cleaning with deionized water.

The baseplate was composed of red copper. Square grooves were formed on the upper surface by machining for vapor flow. There were a total of 11 square grooves with dimensions of $1 \times 1 \times 23$ mm$^3$. A vapor collecting region was set at the end of the grooves and connected to a vapor collector on the upper cover plate to form a complete vapor flow path. The back of the copper baseplate was connected to the heating plate using thermally conductive silicone grease.

The heating plate was a phosphorus-doped silicon plate with a resistance of about 50 $\Omega$ and dimensions of $20 \times 20 \times 0.5$ mm$^3$ The electrodes were affixed to both ends of the heating plate by ultrasonic welding, and connected to a DC power supply to generate joule heat. The voltage at both ends of the electrode could be controlled to limit heating power.

See Table 3 for the range of experimental conditions in this study. The liquid inlet temperature of the compensation chamber was maintained at 10 °C. The heating power was 10–81 W, and the liquid-vapor pressure difference was 0–22 kPa.

**Table 3.** Test conditions.

| Parameter | Unit | Value |
|---|---|---|
| Inlet liquid temperature, $T_{cc}$ | °C | 10 |
| Heat sink temperature, $T_{sink}$ | °C | 5 |
| Heating power, $Q$ | W | 10~80 |
| Heat flux, $q$ | W/cm$^2$ | 2.5~20 |
| Liquid-vapor pressure difference, $\Delta p_{LV}$ | kPa | 0~22 |

### 2.2. Measurement and Uncertainty

Pressure was measured using an absolute pressure transmitter with a range of 0–600 kPa and an accuracy of 0.25 FS%. The pressure measuring points were set on the sidewall of the compensation chamber and the vapor pipeline near the vapor collector. Temperature was measured using a T-type thermocouple with a range of 200–350 °C and an accuracy of ±0.5 °C. The thermocouple had a core diameter of 0.127 mm and a probe tip diameter of 1.5 mm. There were 7 temperature measuring points: at the inlet/outlet of the compensation chamber; the upper surface of the copper baseplate(mini-channels side); the back surface of the copper baseplate (1 mm away from the mini-channels bottom); the outlet of the vapor collector; the liquid tank; the simulated heat source surface. A digital display DC power supply was used to monitor both the current and voltage of the simulated heat source, with a measurement accuracy of current and voltage of 0.01 A and 0.1 V, respectively. Under these conditions, the uncertainties of pressure, liquid-vapor pressure difference, temperature, current, voltage, and power measurement were 0.9%, 1.3%, 3.0%, 3.0%, 0.4%, and 1.8%, respectively.

For indirectly measured parameters, the uncertainties were calculated by a deviation transfer formula [28]:

$$\Delta Y = \sqrt{\sum_{i=1}^{n} \left( \frac{\partial Y}{\partial X_i} \Delta X_i \right)^2} \tag{1}$$

where $Y = f(X_1, X_2, \ldots, X_n)$ is the indirect measured parameter and $X_i$ is the direct measurement parameter.

Hence, the uncertainties of the heat transfer coefficient and thermal resistance calculated by Equations (2) and (3) were 4.0% and 7.5%, respectively.

## 3. Results

### 3.1. Start-Up Characteristics

The start-up characteristics of a pump-assisted capillary phase-change loop under liquid-vapor pressure differences $\Delta p_{LV} = 0, 2, 4, 6, 8$, and 16 kPa are shown in Figure 3a–f, respectively. By comparison, it was found that when $\Delta p_{LV} = 0$ kPa (Figure 3a), the start-up characteristics

of the phase-change loop are different from those at $\Delta p_{LV} > 0$ kPa. With the increase in heating power, the outlet temperature $T_v$ of the vapor collector increased first, was maintained at 65 °C (saturation state), and finally increased to about 80 °C (overheating state). The heat transfer modes of the copper baseplate can be divided into four types: flooded, partially flooded, thin-film evaporation, and overheating according to the change in $T_v$. From the perspective of stable temperature (horizontal section), the heat transfer process was dominated by single-phase convection under low heating power, and the heat transfer efficiency was low with a relatively higher temperature of the copper baseplate $T_{eva}$. At that time, the heat transfer modes were flooded and partially flooded. With the increase in heating power, the heat transfer mode changed to efficient thin-film evaporation. With the increase in heating power, the increased magnitude of $T_{eva}$ decreased. When the heating power exceeded 38 W, the liquid could not completely infiltrate the copper baseplate, part of the vapor channel began to be exposed, and the vapor at the outlet of the vapor collector became overheated. At that point, the heat transfer mode was overheating. As for the time required to reach stable temperature (rise section), it took more than 500 s under flooded and partially flooded modes, but less than 100 s under thin-film evaporation mode.

When the liquid-vapor pressure difference increased to above 2 kPa, the liquid replenishment capacity was enhanced and overheating mode was avoided. Meanwhile, with the increase in liquid-vapor pressure the copper baseplate temperature $T_{eva}$ decreased under the same heating power. In addition, the difference between $T_{eva}$ and the vapor collector temperature $T_v$ under high power decreased, indicating that the liquid-vapor pressure difference could enhance the heat dissipation capacity of the phase-change loop.

Additionally, according to the start-up characteristics of the phase-change loop under different liquid-vapor pressures, the copper baseplate temperature $T_{eva}$ was relatively stable under both low and high power, without temperature fluctuation or temperature overshoot, indicating that the phase-change loop could eliminate two-phase flow instability.

### 3.2. Steady-State Heat Transfer Characteristics

The steady-state heat transfer characteristics directly reflect the heat transfer performance of the phase-change loop. This paper presents a systematic study of the steady-state baseplate temperature, heat transfer coefficient, and thermal resistance of the phase-change loop with the following two influence factors taken into consideration: liquid-vapor pressure difference and heating power.

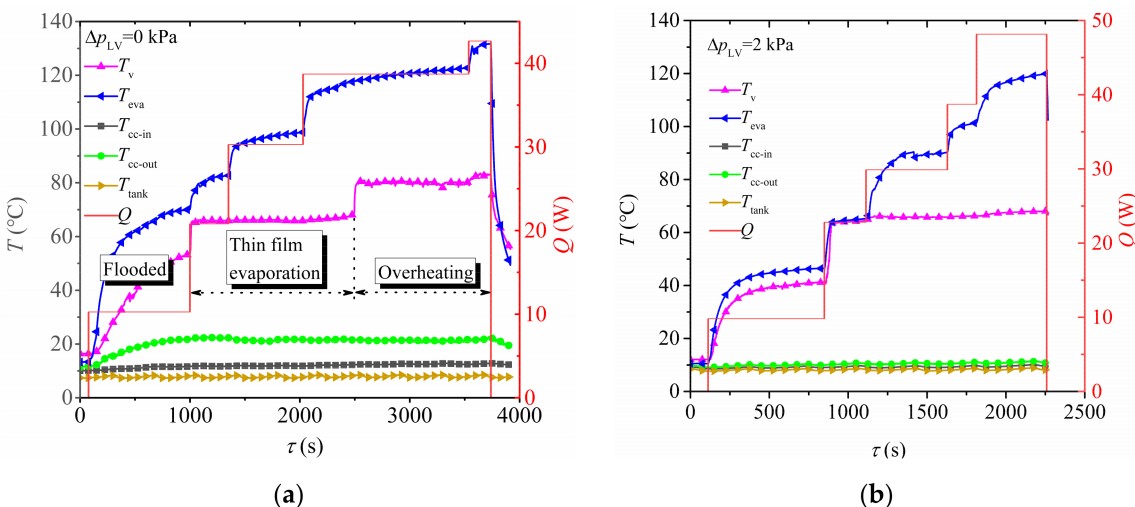

**Figure 3.** *Cont.*

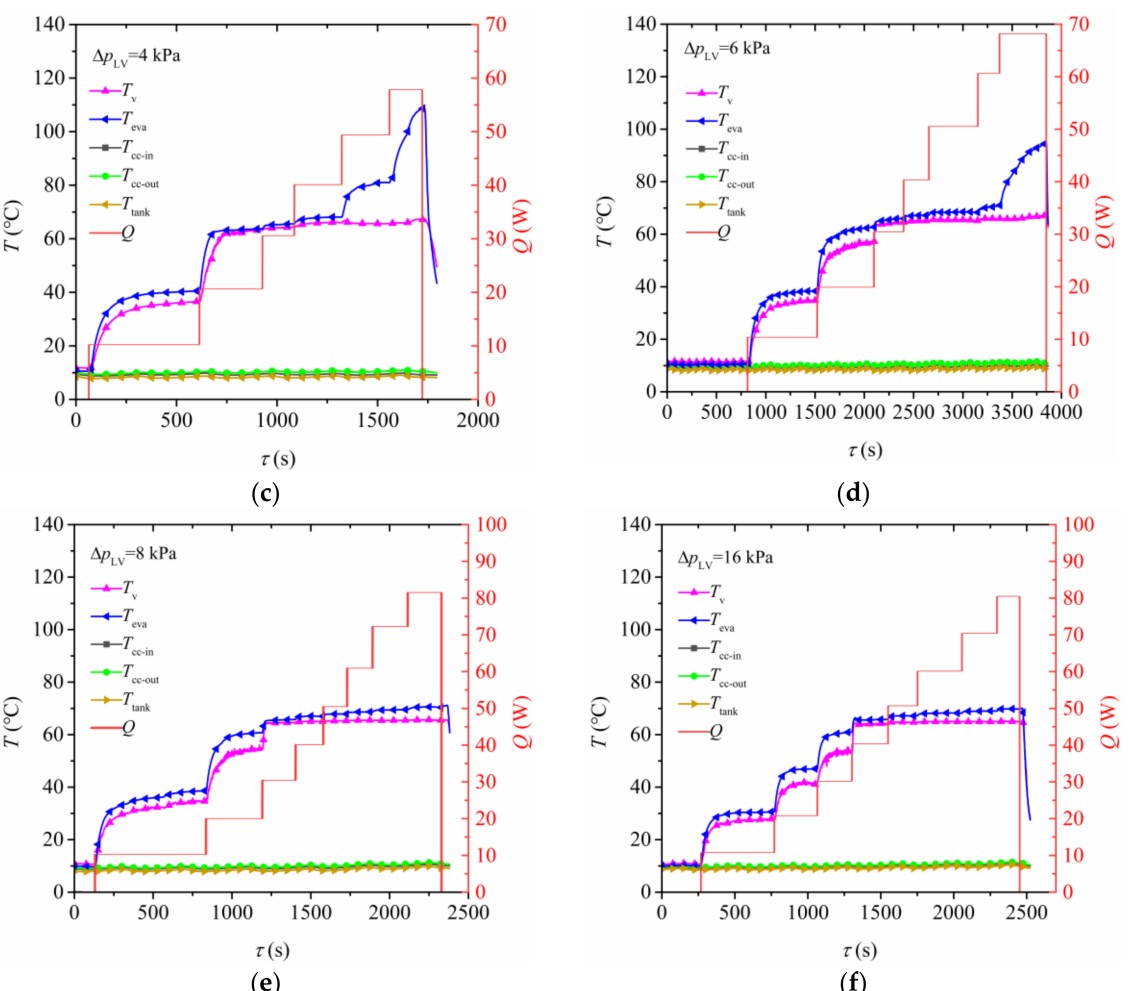

**Figure 3.** Start-up characteristics of the hybrid phase-change loop under different pressure differences: (**a**) $\Delta p_{LV} = 0$ kPa; (**b**) $\Delta p_{LV} = 2$ kPa; (**c**) $\Delta p_{LV} = 4$ ka; (**d**) $\Delta p_{LV} = 6$ kPa; (**e**) $\Delta p_{LV} = 8$ kPa; (**f**) $\Delta p_{LV} = 16$ kPa.

The heat transfer coefficient of the copper baseplate can be calculated according to the following formula:

$$h = \frac{Q}{A(T_{eva} - T_f)} \tag{2}$$

where $Q$ is heating power, $A$ is the heat transfer area equal to the projected area of the baseplate covered by the wick, $T_{eva}$ is the baseplate temperature, and $T_f$ is the fluid temperature at the vapor collector outlet.

The thermal resistance of the phase-change loop can be calculated according to the following formula:

$$R = \frac{T_{eva} - T_{sink}}{Q} \tag{3}$$

where $T_{sink}$ is the outlet fluid temperature of the chiller.

### 3.2.1. Baseplate Temperature

The variation in the copper baseplate temperature $T_{eva}$ in response to changes in liquid-vapor pressure and heating power under steady-state conditions are shown in Figure 4. It can be seen that as long as the liquid-vapor pressure difference was 0 and 2 kPa, a small change in heating power might lead to a rapid rise in $T_{eva}$. However, when the liquid-vapor pressure difference was 4 kPa, there were two obvious turning points in the variation curve of $T_{eva}$ as the heating power increased. After exceeding the first

turning point, $T_{eva}$ changed slightly with the heating power, and the power–temperature curve became steep. When the heating power continued to increase, a second turning point occurred, and $T_{eva}$ began to increase rapidly again. The first turning point occurred after the heat transfer process was significantly enhanced by the heat transfer mode switching from partially flooded to thin-film evaporation. The second turning point occurred after the heat transfer mode switched from thin-film evaporation to overheating. When the liquid-vapor pressure difference exceeded 4 kPa, the power–temperature curve became steep after a turning point. Additionally, with the increase in liquid-vapor pressure difference, the heating power at the turning point also increased gradually.

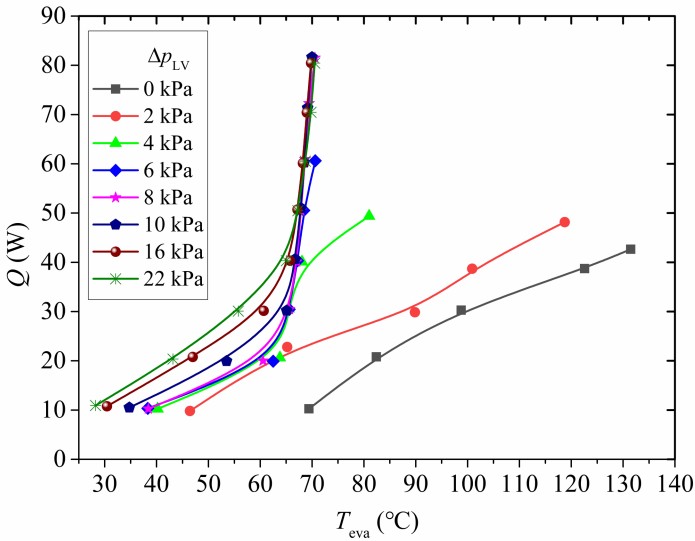

**Figure 4.** Variation in baseplate temperature under different liquid-vapor pressures and heating powers.

However, a high liquid-vapor pressure difference is not always recommended in engineering applications. For example, under current experimental conditions, efficient heat dissipation could be achieved simply by maintaining a liquid-vapor pressure difference lower than 8 kPa. However, if the liquid-vapor pressure difference exceeded 8 kPa, unnecessary pumping power was the result.

### 3.2.2. Heat Transfer Characteristics

The variation in the heat transfer coefficient (HTC) of the evaporator with the liquid-vapor pressure difference and heating power is shown in Figure 5. In Figure 5, the HTC increases first, and then decreases with an increase in heating power under with liquid-vapor pressure differences. Particularly when the liquid-vapor pressure difference was 0 and 4 kPa, the HTC increased first to the maximum and then decreased rapidly. By contrast, when the liquid-vapor pressure difference exceeded 8 kPa, the HTC decreased gradually after reaching the maximum. Regardless, the change in HTC was closely related to the transition of heat transfer modes. Under low liquid-vapor pressure difference, with the increase in heating power, the heat transfer mode switched from flooded and partially flooded to thin-film evaporation, resulting in a rapid increase in HTC. If the heating power continued to increase, the heat transfer mode switched from thin-film evaporation to overheating, leading to a decrease in the HTC. However, under a high liquid-vapor pressure difference, there would be no transition from thin-film evaporation to overheating at the current heating power. Therefore, the HTC would not decrease rapidly.

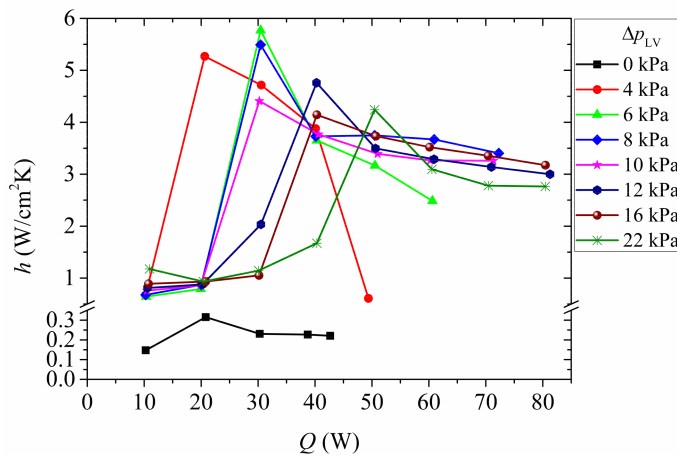

**Figure 5.** Variation in heat transfer coefficient under different liquid-vapor pressure difference and heating powers.

In summary, when the liquid-vapor pressure difference was low, the range in power required to maintain high HTC was small and overheating mode was easily entered. In contrast, a high liquid-vapor pressure difference could ensure that a high heat transfer coefficient could be maintained in a wider power range. Under a higher liquid-vapor pressure difference, the HTC would increase rapidly with heating power. In engineering applications, the liquid-vapor pressure difference could be actively adjusted according to the operation characteristics of electronic devices in order to achieve the optimum heat dissipation effect.

### 3.2.3. Regime Map of Different Heat Transfer Modes

The heat transfer modes of the baseplate fundamentally depended on the balance between evaporation and liquid replenishment. The evaporation was related to heating power, while the liquid replenishment was related to the driving force (liquid-vapor pressure difference and capillary force). When the liquid replenishment was much greater than the evaporation, the heat transfer process was dominated by convection, leading to flooded or partially flooded states. However, when the liquid replenishment slightly exceeded the evaporation, a stable evaporating meniscus could be maintained within the vapor channel and the liquid could completely wet the vapor channel, leading to thin-film evaporation mode. When the liquid replenishment was slightly lower than the evaporation, the liquid bridge between the liquid and capillary wick in the vapor channel would disappear, and a part of the vapor channel would expose and overheat the generated vapor. At that time, the heat transfer mode was overheating.

From the above discussion, thin-film evaporation mode is superior to the other three heat transfer modes in heat transfer efficiency and start-up speed, and is also the most desirable state in engineering applications. As for the copper baseplate, its heat transfer mode is closely related to the liquid-vapor pressure difference and heating power. A two-dimensional heat transfer mode distribution diagram recording both liquid-vapor pressure variation and heating power was drawn after a number of experiments, as shown in Figure 6. Under a high liquid-vapor pressure difference or low heating power, the liquid replenishment capacity was high, the heat transfer was dominated by single-phase convection, and the heat transfer modes were mainly flooded and partially flooded. Under low liquid-vapor pressure or high heating power, the liquid replenishment capacity was low while the evaporation rate was high and the heat transfer mode would switch to overheating. The thin-film evaporation mode lay between the two, with its distribution area in the shape of a horn. In engineering applications, the liquid-vapor pressure difference could be regulated dynamically according to the heat transfer mode distribution diagram in order to maintain an efficient thin-film evaporation mode at a certain heating power.

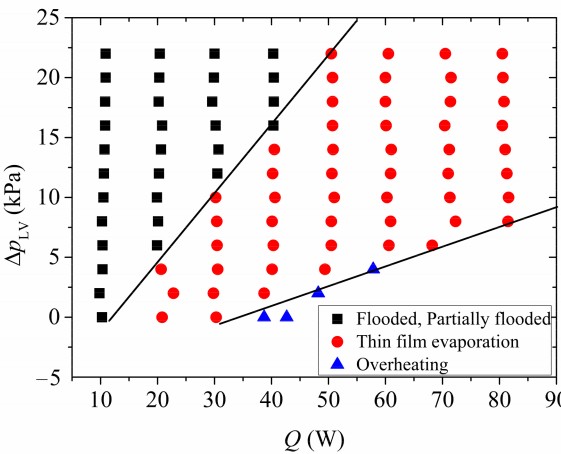

**Figure 6.** Two-dimensional heat transfer mode distribution diagram.

The variation of in resistance of the phase-change loop with liquid-vapor pressure and heating power differences is shown in Figure 7. The thermal resistance decreased with the increase in heating power under various pressure differences. When the power was below 40 W, thermal resistance constantly decreased with the increase in liquid-vapor pressure. For example, when the liquid-vapor resistance decreased from 6.0 K/W to 1.8 K/W, a decrease of 70%, which indicates that heat transfer could be significantly enhanced by increasing the liquid-vapor pressure difference at lower power. When the power was higher than 40 W and the liquid-vapor pressure difference exceeded 8 kPa and the thermal resistance curves fairly coincided, indicating that the thermal resistance could not be reduced at all times by increasing the liquid-vapor pressure under high power.

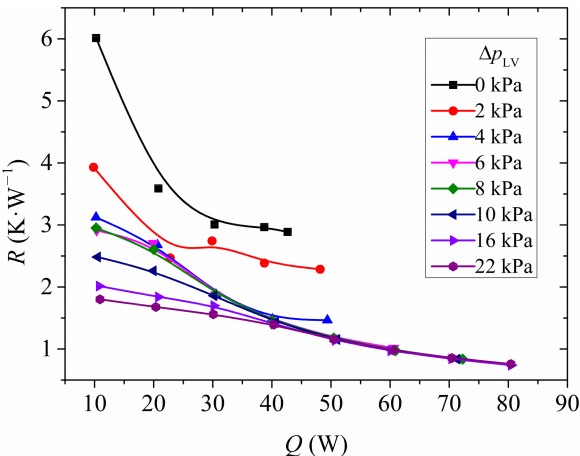

**Figure 7.** Variation in thermal resistance under different liquid-vapor pressure differences and heating powers.

## 4. Conclusions

In order to solve the problem encountered by the traditional boiling heat dissipation technology, i.e., two-phase flow instability, a pump-assisted capillary phase-change loop with methanol as the working medium was established. The effect of liquid-vapor pressure difference and heating power on the start-up characteristics and steady-state heat transfer characteristics was studied. The conclusions are as follows:

(1) There was no temperature fluctuation in the starting process under various liquid-vapor pressure differences and powers, indicating that the pump-assisted capillary phase-change loop overcame the two-phase flow instability. There were four modes

of heat transfer, including flooded, partially flooded, thin-film evaporation, and overheating. It took the shortest time (i.e., within 100 s) to start in thin-film evaporation mode, while it took more than 500 s in the other three modes.

(2) Both liquid-vapor pressure difference and power were not only important to the heat transfer modes, but also determined the steady-state heat transfer performance of the phase-change loop. The heat transfer coefficient could be significantly improved and the thermal resistance reduced by increasing liquid-vapor pressure as long as it did not exceed 8 kPa. For instance, if the liquid-vapor pressure difference increased from 0 to 8 kPa while the heating power remained at 30 W, the heat transfer coefficient could be improved by nearly 27 times and the thermal resistance reduced by 37%. When the liquid-vapor pressure difference exceeded 8 kPa, the temperature power curve and thermal resistance power curve fairly coincided under high heating power. At that time, if the liquid-vapor pressure differential continued to increase, the heat transfer enhancement effect decreased and more pumping power resulted.

(3) The thin-film evaporation mode with the maximum heat transfer efficiency was the most desirable heat transfer state in engineering applications. A two-dimensional heat transfer mode distribution diagram depicting both liquid-vapor pressure difference and heating power was created after a number of experiments. The thin-film evaporation mode was between flooded/partially flooded and overheating in the shape of a horn. During engineering applications, the difference in liquid-vapor pressure could be controlled to maintain efficient thin-film evaporation at the baseplate of the evaporator to achieve the optimum heat dissipation effect.

(4) The capillary wick is a core part of the evaporator. Therefore, capillary wick material and its primary performance parameters (pore size, porosity, and permeability), as well as the type of its adhesion to the baseplate—which has an important effect on the heat transfer performance of the phase change loop—deserve in-depth and systematic research in the future.

**Author Contributions:** Writing—original draft preparation, X.Y. and G.W.; writing—review and editing, C.Z.; investigation, data curation, J.L.; conceptualization, project administration, funding acquisition, X.Y. and J.W. All authors have read and agreed to the published version of the manuscript.

**Funding:** The research was supported by the National Natural Science Foundation of China, grant numbers 52006166 and 51961135102; and the Fundamental Research Funds for the Central Universities, grant number XJH012020040.

**Conflicts of Interest:** We declare that we have no financial and personal relationships with other people or organizations that can inappropriately influence our work, there is no conflict of interest in the manuscript entitled "Experimental study on the heat transfer performance of pump-assisted capillary phase-change loop".

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
