# Peer review of "Experimental Study on the Heat Transfer Performance of Pump-Assisted Capillary Phase-Change Loop"

_applsci, doi:10.3390/app112210954_

Round 1

Reviewer 1 Report

dear Authors,

Nice paper, well organized and written.

Some minor comments:

1) state of the art into the introduction: strange to not find any paper by Prof. J. R. Thome from EPFL, since it has the largest scientific production concerning the heat sinks

2) what is the definition of Dplv? it looks like you are able to control it but it is not defined approprieately.

Also, Do you mean as DPlv an average value during the experiment? or a precise value kept constant as a boundary condition? In the second case more details about the experimental procedure are needed.

3) what is the definition of A, heat transfer area and how do you measure it?

4)which is the uncertainty in the measurement of the heat transfer coefficient?

Author Response

Some minor comments:

1)State of the art into the introduction: strange to not find any paper by Prof. J. R. Thome from EPFL, since it has the largest scientific production concerning the heat sinks.

Response: Thanks very much for the reviewer’s suggestion. The related papers of Prof. J. R. Thome has been added. (See: line 48 in Page 2 and lines 402-409 in Page 12 in the Marked Revision)

2)What is the definition of Dplv? it looks like you are able to control it but it is not defined approprieately. Also, Do you mean as DPlv an average value during the experiment? or a precise value kept constant as a boundary condition?

Response: Thanks very much for your comments. The ΔPLV is defined as the pressure difference between the liquid pressure in compensation chamber and the outlet pressure of vapor collector, which is described in the manuscript in lines 112-118 in Page 3 in the Marked Revision.

During the experiment, the value of ΔPLV is kept constant as a boundary condition by adjusting the liquid pressure PL in the compensation. Meanwhile, the hot fluid pressure PL is affected by heating power.

3)In the second case more details about the experimental procedure are needed.

Response: Thanks very much for your comments. The details about the experimental procedure are added. (See: lines 120-127 in Page 3 in the Marked Revision)

4)What is the definition of A, heat transfer area and how do you measure it?

Response: Thanks very much for your comments. A is equal to the projected area of the baseplate covered by the wick, A=23*23 m2 in this work. (See: Table 2 in Page 5 and lines 232-233 in Page 8 in the Marked Revision)

5) Which is the uncertainty in the measurement of the heat transfer coefficient?

Response: Thanks very much for your comments. The uncertainties of heat transfer coefficient and thermal resistance are 4.0% and 7.5%, respectively. The calculating method is shown in lines 181-187 in Pages 5-6 in Marked Revision.

Reviewer 2 Report

The work is interesting and should be published.

Figure 6 shows an idea of the flow patterns of liquid and vapor in different boiling regimes.

However, no visualization tests were performed in the work. The presented concept of patterns seems realistic, however, in my opinion, the work should not contain diagrams of patterns that have not been observed. Authors should limit themselves to discussing the results only.

Minor comments:

There is no information about the size of the thermocouple and no information about the distance from the mini-channels bottom.

In Figure 3 the lines are marked with colors - in my opinion a different marker should be used.

Author Response

The work is interesting and should be published.

1)Figure 6 shows an idea of the flow patterns of liquid and vapor in different boiling regimes. However, no visualization tests were performed in the work. The presented concept of patterns seems realistic, however, in my opinion, the work should not contain diagrams of patterns that have not been observed. Authors should limit themselves to discussing the results only.

Response: Thanks very much for the reviewer’s suggestion. According to the reviewer’s comments, we have deleted Fig.6 to make the expressions to be more rigorous.

Minor comments:

2)There is no information about the size of the thermocouple and no information about the distance from the mini-channels bottom.

Response: Thanks very much for the reviewer’s comment. The thermocouple has a core diameter of 0.127 mm and probe tip diameter of 1.5 mm. One thermocouple is adhered to the upper surface of the copper baseplate (mini-channels side) to measure the surface temperature, i.e Teva. One thermocouple is adhered to the back side of the copper baseplate, which is 1 mm away from the mini-channels bottom. (See: lines 171-175 in Page 5)

3)In Figure 3 the lines are marked with colors - in my opinion a different marker should be used.

Response: Thanks very much for the reviewer’s comment. The lines in Fig.3 are marked by symbols instead of colors in Marked Revision.
